# Prevalence and characterisation of carbapenemase encoding genes in multidrug-resistant Gram-negative bacilli

**Sayran Hamad Haji** [1]*, **Safaa Toma Hanna Aka** [1], **Fattma A. Ali** [2]

**1** Department of Pharmacognosy, College of Pharmacy, Hawler Medical University, Erbil, Iraq, **2** Department of Medical Microbiology, College of Health Science, Hawler Medical University, Erbil, Iraq

* sayran.haji@hmu.edu.krd

**Data Availability Statement:** The minimal dataset underlying this study is available on Mendeley (https://data.mendeley.com/datasets/dp7g8rntyb/1).

## Abstract

### Background

Emerging worldwide in the past decade, there has been a significant increase in multidrug-resistant bacteria from serious nosocomial infections, especially carbapenemase-producing Gram-negative bacilli that have emerged worldwide. The objective of this study is to investigate carbapenem resistance in Gram-negative bacilli bacteria using phenotypic detection, antimicrobial resistance profiles and genotypic characterisation methods.

### Methods

200 Gram-negative bacilli isolates were collected from different clinical specimens. All clinical samples were exposed to isolation and identification of significant pathogens applying bacteriological examination and an automated Vitek-2 system. The isolates were subjected to susceptibility tests by the Vitek-2 automated system and those isolates that were resistant to beta-lactam drugs, including carbapenems, third-generation cephalosporines or cefoxitin, were selected for phenotyping using Carba plus disc system assay for detection of carbapenemase-producing isolates. These isolates were further confirmed by molecular detection. PCR was used for the detection carbapenem-resistant genes (OXA-48, IMP, NDM, VIM, and KPC).

### Results

110 (55%) of 200 Gram-negative bacilli were identified as beta-lactam-resistant isolates. The frequency of carbapenem-resistant isolates was calculated to be 30.9% (n = 34/110). A collection totalling 65/110 (59%) isolates were identified as carbapenemase producers by phenotypic method. Moreover, among the 65 carbapenemase-producing Gram-negative isolates with a positive phenotype-based result, 30 (46%), 20 (30%) and 18 (27%) isolates were positive for OXA-48, KPC and MBL enzymes, respectively, as well as the production of 27% of AmpC with porin loss. Tigecycline was the most effective antibiotic that affected 70% of MDR isolates, but high rates of resistance were detected to other tested antimicrobials. Of interest, a high incidence of MDR, XDR and PDR profiles were observed among all

**Funding:** The author(s) received no specific funding for this work.

**Competing interests:** The authors have declared that no competing interests exist.

carbapenemase-producing isolates. 36% (24/65) of the tested isolates were MDR to 3 to 5 antimicrobial classes. 29% (17/65) of the recovered isolates were XDR to 6 to 7 antimicrobial classes. Alarmingly, 24% (16/65) of isolates displayed PDR to all the tested 8 antimicrobial classes. Genotype assay, including 53 phenotypically confirmed carbapenemase-producing isolates of Gram-negative bacilli, found 51(96%) isolates were harbouring one or more genes. The most common carbapenemase gene was $bla_{NDM}$ 83% (44/53) followed by $bla_{OXA-48}$ 75% (40/53), $bla_{VIM}$ 49% (26/53) and $bla_{IMP}$ 43% (23/53), while the gene $bla_{KPC}$ was least frequent 7% (4/53). 92% (46/51) of isolates were involved in the production of more than one carbapenemase gene.

## Conclusion

This study demonstrated the emergence of carbapenemase-producing Gram-negative pathogens implicated in healthcare-related infections. Accurate identification of carbapenem-resistant bacterial pathogens is essential for patient treatment, as well as the development of appropriate contamination control measures to limit the rapid spread of pathogens. Tigecycline exhibited potent antimicrobial activity against MDR, XDR and PDR-producing strains that establish a threatening alert which indicates the complex therapy of infections caused by these pathogens.

## Introduction

Multidrug resistance has increased globally and is considered a public health threat. Several recent studies have reported the existence of multidrug-resistant bacterial pathogens from different origins including humans, poultry, cattle and fish. This has increased the need for routine application of antimicrobial susceptibility testing to detect the antibiotic of choice as well as screening of the emerging MDR strains [1, 2].

In general, the phenomenon of MDR is primarily attributed to the recurrent and indiscriminate use of antibiotics as well as coding for some antimicrobial resistance genes [3].

In the last decade, an alarming increase in the prevalence of multidrug-resistant bacteria of serious nosocomial infections, especially carbapenemase-producing Gram-negative bacilli has been shown worldwide [4, 5]. Carbapenems (i.e. meropenem, imipenem, ertapenem and doripenem) are a safe and highly effective class of antibiotics and considered last line drugs for controlling multidrug-resistant Gram-negative pathogens including extended-spectrum beta-lactamases-producing *Enterobacteriaceae* [6]. Owing to their broad-spectrum activity, carbapenems are frequently used in the treatment of life-threatening infections. Over-prescribing of these drugs has increased carbapenem resistance. This problem is one of the main reasons for the expression of carbapenemase genes among members of this family [7]. Carbapenemase enzymes are the most common mechanism by which resistance to carbapenems arises [8, 9]. Carbapenem resistance has been labelled in *Enterobacteriaceae*, mostly in *Klebsiella pneumoniae* compared to *Escherichia coli* or other Enterobacterial species, and in non-fermentative Gram-negative bacilli such as *Pseudomonas aeruginosa* and *Acinetobacter baumannii* [4]. These Gram-negative bacteria are the main nosocomial pathogens and the most prevalent bacteria responsible for a range of hospital-acquired and community-acquired difficult to treat infections such as meningitis, pneumonia, peritonitis, septicaemia and urinary tract infections.

Infection acquired by carbapenemase-producing Gram-negative bacilli is associated with high in-hospital mortality (up to 70%) due to the inefficiency of initial treatment regimens [10].

Clinically relevant Gram-negative bacilli infections include a variety of species. Urinary tract infections (*Escherichia coli*, *Enterobacter species*, *Proteus species*, *Serratia marcescens*), gastrointestinal infections (*Salmonella typhi*, *Shigella sp.*, *Salmonella enteritidis*, *Helicobacter pylori*) and respiratory tract infections (*Pseudomonas aeruginosa*, *Klebsiella species*, *Haemophilus*, *Legionella pneumophila*) are Gram-negative primary infections in most hospitals [11].

Recently, a non-fermenting Gram-negative bacterium, *Acinetobacter baumannii*, has been linked with nosocomial infections. Likewise, other Gram-negative organisms such as *Aeromonas sp.* and *Stenotrophomonas maltophilia* have been increasingly reported in many incidents of bacteraemia, pneumonia, meningitis, and surgical or wound site infections in hospital intensive care settings. Most of these Gram-negative microorganisms are resistant to nearly all antimicrobial drugs available worldwide [12].

Various mechanisms are found, for instance acquisition of genes encoding beta-lactamases that destroy the antibiotic, efflux pumps to eliminate agents from the bacterial cell before the target site is reached, and modification of binding sites that may occur before adequate concentrations are reached to remove the whole polymicrobial community. Gram-negative organisms are no exception to these interactions [13].

Carbapenems have become the last resort antibiotics in the treatment of many serious bacterial infections, particularly in hospital settings. However, most drugs, whether beta-lactam or other non-beta-lactam antimicrobials have been ineffective against multi-resistant Gram-negative bacteria [12]. Therefore, acquired carbapenemase-producing Gram-negative bacilli are an actual clinical concern for antimicrobial administration. Specifically those resistant to carbapenems may be associated with resistance to other classes of antibiotics, such as fluoroquinolones and aminoglycosides [4].

Initially, carbapenemases were chromosomally mediated in a few specific species, but now they mediate a plasmid, or both a chromosomal and a plasmid. This results in a more violent spread of resistance due to horizontal transfer between different bacterial species and genera [9]. Since 1993, different types of carbapenemases have been recognised belonging to three molecular classes, the Ambler classes A, B and D beta-lactamases [14]: class A carbapenemases, *Klebsiella pneumoniae* carbapenemase (KPC) enzymes; class B metallo-beta-lactamases (MBLs) such as New Delhi metallo-beta-lactamases (NDM), Verona integron-encoded metallo-beta-lactamase (VIM) and imipenemase (IMP) type enzymes; and class D carbapenem-hydrolysing oxacillinase (OXA) such as OXA-48 enzymes [15].

Class A beta-lactamase has the ability to degrade penicillins, carbapenems and cephalosporins. Class B metallo-beta-lactamases exhibit different hydrolysis activities against all beta-lactams except for monobactam (aztreonam). Their activity is inhibited by EDTA but not by clavulanic acid [14, 16]. The OXA-48 enzyme shows low activity against carbapenems, high activity against penicillin but only low activity against 3rd and 4th generation cephalosporins. However, this is rarely a treatment option because other beta-lactamases, such as ESBL, are often associated [9]. Globally, most common carbapenemases consist of MBLs (IMP, VIM and NDM), OXA and KPC. The production of carbapenemases including OXA-48, NDM and KPC are the predominant resistance mechanisms among carbapenem-resistant *Enterobacteriaceae* clinical isolates [17]. In addition, the production of carbapenemases may be related to their hydrolytic action and levels of resistance to broad-spectrum beta-lactams, along with the potential migration of these genes [10].

Different types of mechanisms can mediate carbapenem resistance among the non-carbapenemase-producing or carbapenemase-producing Gram-negative microorganisms. Carbapenemases are beta-lactamases enzymes that are capable of actively cleaving the amide bond in

beta-lactam rings of nearly all beta-lactam antibiotics, including the carbapenems, rendering them inactive [16, 18]. The non-carbapenemase-producing carbapenem-resistant microorganisms produce other mechanisms of resistance that can be related to porin mutation in the outer membrane or the presence of an efflux pump, or both, with the production of ESBL or AmpC beta-lactamase dependent on the specific Gram-negative microorganism. All of the carbapenem-resistant microorganisms are of concern, as they are likely to be multidrug-resistant [4, 18].

Nevertheless, carbapenemase-producing microorganisms are of most concern and therefore targeted by antibiotic stewardship and infection control programs, as it is believed to be the primary mechanism responsible for the spread of carbapenem resistance among Gram-negative microorganisms globally [18]. Carbapenem-resistant *Enterobacteriaceae* are highly lethal and treatment choices are limited [19]. Since in one study the mortality rate was 22%, it was necessary to assess the propensity for carbapenem resistance in hospitals in Erbil city, Iraqi Kurdistan Region, and the genetic characteristics responsible for resistance [20].

Practical and rapid detection of carbapenemases is essential for targeted antimicrobial therapy and to ensure early control of infection in hospitals. The detection of carbapenemases and the accurate and rapid identification of their genes in *Enterobacteriaceae* remains a major public health challenge for clinical laboratories [10]. The MASTDISCS combi Carba plus disc system (MAST-Carba plus, Mast Group Ltd., UK), is an initial phenotypic assay and is a practical and fast method for distinguishing carbapenemase activity regardless of carbapenemase genes from *Enterobacteriaceae* in the clinical laboratory. This method is based on the use of a combination of faropenem (disc A) and MBLs (disc B), KPC inhibitor (disc C), AmpC inhibitor (disc D), and temocillin in combination with MBLs inhibitor (disc E). These have recently been described as promising methods for the detection of organisms with MBLs, KPC and OXA-48 enzymes [21]. Molecular techniques remain the gold standard for identifying and characterising states of carbapenemases. The most common of them is polymerase chain reactions [9].

The polymerase chain reaction is a specific and highly dependable diagnostic instrument that simplifies the detection of pathogenic microorganisms, virulence genes as well as antimicrobial resistance genes [22].

The overall objective of the study is to investigate the incidence of carbapenem resistance in Gram-negative bacilli pathogens isolated in hospitals in Erbil city, Iraqi Kurdistan Region. This process was conducted using phenotype-based detection, along with their antimicrobial resistance profiles to different classes of antibiotics, to characterise the prevalence of carbapenemase resistance genes (KPC, IMP, NDM, VIM, OXA-48) using polymerase chain reaction (PCR).

## Materials and methods

A total of 200 cultures with positive growth of Gram-negative bacilli isolates (i.e. non-fermenter and *Enterobacteriaceae*) were collected from different clinical specimens. All of the clinical samples were exposed to isolation and identification of significant pathogens using bacteriological investigation, including microscopy of the Gram-stained preparations and biochemical analysis using Vitek-2 automated system with the AST-GN card (bioMerieux, USA) according to the manufacturer's instructions from the microbiology laboratory at hospitals. Bacteria were stored from single colonies in trypticase soy broth with 40% glycerol at -70˚C until used [23].

These isolates were subjected to susceptibility tests by the Vitek-2 automated system with the AST-GN card (bioMerieux, USA) and those that were resistant to broad-spectrum beta-

lactam drugs including carbapenems, third-generation cephalosporines or cefoxitin, were selected for further characterisation. 110/200 (55%) beta-lactam-resistant isolates of Gram-negative bacilli were recovered from different clinical specimens: urine 69 (62%), sputum 18 (16%), swab 16 (14%) and blood 7 (6%). The frequency of carbapenem-resistant isolates was calculated to be 30.9% (n = 34). A collection totalling 65/110 (59%) were identified as carbapenemase-producing isolates by the phenotype-based method using the Carba plus disc system assay, and further confirmed by molecular detection. The median age of the patients was 40 years, with a range of 1–80 years. Among 65 patients, 39 (60%) were female. The bacterial strains were isolated from a variety of infection sites including urine (38), sputum (15), wound swab (8) and blood (4). The isolates were collected from community and hospitalised patients at hospitals in Erbil city, Iraqi Kurdistan Region, during 2019–2020.

## Antibacterial susceptibility test of carbapenemase-producing Gram-negative bacilli

In the current study, the following antimicrobials were tested: piperacillin and piperacillin-tazobactam (β-lactams and β-lactamase-inhibitor combination); ceftazidime, ceftriaxone, cefazoline, cefepime and cefoxitin (cephalosporins); gentamicin and amikacin (aminoglycosides); imipenem, ertapenem and meropenem (carbapenems); ciprofloxacin (quinolone); trimethoprim/sulphamethoxazole (sulphonamide); tigecycline (tetracycline); and nitrofurantoin (in urinary infection). They were determined for all these isolates using the Vitek-2 automated (GN-20 and 22 AST card) system (bioMerieux, USA) according to the manufacturer's instructions [10]. The resistance profiles were categorised into MDR, XDR and PDR according to Algammal [1].

## Phenotype detection for carbapenemase production using commercial combination disc assay

Carbapenemase production was examined phenotypically for all strains that showed resistance or intermediate resistance to carbapenem drugs, or demonstrated resistance to either third-generation cephalosporins or cefoxitin, using the MASTDISCS® Combi Carba plus (Carba plus) commercial combination disc assay, according to the manufacturer's instructions [10, 21].

MASTDISCS® Combi Carba plus (MASTDISCS combi-D73C, Bootle, UK) is a combined-disc test of a five-disc system: penem (disc A), penem+MBLs inhibitor (disc B), penem+KPC inhibitor (disc C), penem+AmpC inhibitor (disc D) and temocillin+MBLs inhibitor (disc E). Evaluation was to detect KPC, MBL and OXA-48-like carbapenemases formed by *Enterobacteriaceae*, including the reliable differentiation of KPC from AmpC-producing isolates. Adding the temocillin disc in combination with MBL inhibitor (E disc), instead of just the temocillin disc, improves the discrimination of OXA-48 by removing the confusion of MBLs incorrectly identified as OXA-48. MASTDISCS® Combi Carba plus D73C was performed according to the manufacturer's instructions. First, a 0.5 McFarland inoculum of the test organism was prepared and uniformly spread across the surface of Mueller–Hinton agar plates. Five prepared discs were placed onto the inoculated medium by sterile forceps, ensuring sufficient space between discs to allow obviously clear inhibition zones to be formed. After 18 to 24 hours of incubation at 37˚C, the diameter of any zones of inhibition observed were measured and recorded in millimetres (mm) disregarding any small colonies within the region. The results were interpreted according to the manufacturer's instructions [21]: discs showing no zone of inhibition were recorded as 6 mm, ≥5 mm difference in the disc zone diameters observed only between disc B and disc A (less than 5 mm difference between disc C or disc D

and disc A) indicated the production of MBLs. Likewise, $\geq 5$ mm difference only between disc C and disc A (less than 5 mm difference between disc B or disc D and disc A) indicated KPC production. AmpC production with porin loss is indicated when both discs C and D show a zone difference of $\geq 5$ mm compared to disc A. In the absence of such synergistic effects among discs A–D, the diameter of the inhibition zone of $\leq 10$ mm around disc E indicated OXA-48 carbapenemase production [10]. *K. pneumoniae* NCTC 13438 (KPC positive), *K. pneumoniae* NCTC 13440 (MBL positive) and *K. pneumoniae* NCTC 13442 (OXA-48 positive) were used as positive controls, and *E. coli* ATCC25922 as negative controls.

## PCR-based detection of carbapenemase genes

53 isolates screened as carbapenemase-positive by the phenotype-based method were further investigated for the detection of carbapenemase genes using PCR-based methods. Singleplex and multiplex PCR were carried out to examine for the presence of five predominant carbapenemase genes namely KPC ($bla_{KPC}$ gene), MBLs ($bla_{IMP}$, $bla_{VIM,}$ and $bla_{NDM}$ genes) and $_{OXA-48}$ ($bla_{OXA-48}$ gene) using a panel of primers as previously described [7, 24, 25]. Multiplex PCR for $bla_{IMP}$ and $bla_{VIM}$ detection was planned. Screening of isolates containing the $bla_{NDM}$, $bla_{OXA-48}$ and $bla_{KPC}$ genes was tested by singleplex PCR. Extraction of the total DNA from a bacterial culture in the logarithmic phase was conducted with the commercial genomic DNA extraction kit (DNALandScientfic Cat No. GG2001) according to the manufacturer's instructions. All the strains were identified for the mentioned genes using the PCR technique through amplification of the gene-specific targeted sequence in a thermocycler machine (Techne/UK).

In singleplex PCR ($bla_{OXA-48}$, $bla_{NDM}$ and $bla_{KPC}$ genes) the premix was subjected in the total volume of 25μl, 12.5 μl Gotaq Green Master Mix (Promega/USA), 3μl of genomic DNA, 1.5 μl for each primer and the volume completed with 6.5μl nuclease-free water. The PCR for $bla_{OXA-48}$ and $bla_{NDM}$ genes were conducted according to the following conditions, initial denaturation 95˚C for 5min, 35 cycles amplification for denaturation at 95˚C for 30 sec, annealing at 56˚C for 1 min, extension step at 72˚C for 1 min then the final extension at 72˚C for 5 min. The multiplex master mix was prepared in total volume of 25μl, 12.5 μl Gotaq Green Master Mix (Promega/USA), 3μl of genomic DNA, 1 μl for each primer and the volume completed with 5.5μl DNase, RNase free water [25].

The Multiplex PCR program reaction for $bla_{IMP}$ and $bla_{VIM}$ were conducted as the following: initial denaturation at 94˚C for 8 min; denaturation at 94˚C for 30 sec, annealing at 54˚C for 55 sec and extension at 72˚C for 1 min; and the final extension at 72˚C for 10 min. The PCR product amplicons were then analysed by agarose gel electrophoresis (2%) and the DNA stained with safe dye (Bioland/USA). The presence and absence of the carbapenemase genes were determined using a UV transilluminator (Syngene/UK) and the PCR product band for all genes identified on the gel as compared to the standard DNA ladder/1kb (Norgenbiotek/Canada).

## Statistical analysis

GraphPad Prism (version 5; GraphPad Software, San Diego, CA). Chi-square test was applied to analyse the collected data. A P-value of < 0.05 was considered statistically significant.

## Ethical issues

This study was approved by the ethics committee for the College of Pharmacy, Hawler Medical University. During data collection, all patients who attended treatment were asked to obtain verbal consent. The purpose of the study was explained to the participants and details about

anonymity and the right to withdraw from the study at any stage were provided. Finally, verbal consent was obtained from the participants in this research.

## Results

### Demographic factors and characteristics of bacterial isolates

A collection of 200 Gram-negative bacilli were collected from different clinical specimens. 110 (55%) broad-spectrum beta-lactam resistant isolates of Gram-negative bacilli were multidrug-resistant (MDR) to different classes of antimicrobials tested, including carbapenems. The isolates comprised *Escherichia coli* (*E. coli*) (n = 49, 44%), *Klebsiella sp.* (*K. pneumoniae* and *K. oxytoca*) (n = 23, 20%), *Acinetobacter baumanii* (*A. baumanii*) (n = 13, 11%), *Pseudomonas aeruginosa* (*P. aeruginosa*) (n = 10, 9%), *Proteus sp.* (*P. vulgaris* and *P. mirablis*) (n = 5, 4%) and *Serratia marcescens* (n = 3, 2%). Less prevalent microorganisms were, *Enterobacter cloacae*, *Stenotrophomonas maltophilia*, *Morganella morganii*, *Achromobacter dentrificans*, *Ewingella americana*, *Sphingomonas sp.* and *Pantoea agglomerans* and included one isolate each (n = 1, 0.9%). The frequency of carbapenem-resistant isolates was calculated to be 30.9% (n = 34). The uppermost resistance isolates to carbapenems were detected for *Acinetobacter baumanii* at 9, while *Klebsiella sp.*, *E. coli* and *Pseudomonas aeruginosa* ranked second in terms of gaining resistance to imipenem with a frequency of 6 for each. Urine samples were found to be the dominant type (62%), followed by sputum (16%). Statistical analysis showed that there was a significant difference in the prevalence of different bacterial pathogens and carbapenemase-resistant isolates recovered from different types of samples (p < 0.05) (Table 1).

All cases of resistance to carbapenems and second or third-generation cephalosporins were first tested by a phenotype-based method using Carba plus test, 65/110 were identified as carbapenemase-producing isolates. These isolates belonged to different species: *E. coli* (n = 20, 30.7%); *Klebsiella sp.* (12 *K. pneumoniae* and 3 *K. oxytoca*), (n = 15, 23%) and (n = 3, 4.6%); *Acinetobacter baumanii* (n = 11, 16.9%); *Pseudomonas aeruginosa* (n = 8, 12.3%); and *Proteus sp.* (*P. vulgaris* and *P. mirablis*) (n = 4, 6.1%). The less predominant microorganisms were,

**Table 1. Prevalence of broad-spectrum beta-lactam resistant Gram-negative bacilli isolates and isolation of carbapenem-resistant isolates from different types of clinical samples.**

| Bacteria | No. of isolates (%) | Carbapenem resistant | Specimen no. and types | | | |
|---|---|---|---|---|---|---|
| | | | Urine | Sputum | Swab | Blood |
| *E. coli* | 49 (44) | 6 | 42 | _ | 3 | 4 |
| *Klebsiella sp.* | 23 (20) | 6 | 13 | 5 | 6 | _ |
| *Acinetobacter baumanii* | 13 (11) | 9 | 1 | 9 | 2 | 1 |
| *Pseudomonas aeruginosa* | 10 (9) | 6 | 7 | 1 | 2 | _ |
| *Proteus sp.* | 5 (4) | 3 | 2 | 2 | 1 | _ |
| *Serratia marcescens* | 3 (2) | 1 | 2 | 1 | | _ |
| *Enterobacter cloacae* | 1 (0.9) | _ | _ | _ | 1 | _ |
| *Stenotrophomonas maltophilia* | 1 (0.9) | 1 | _ | _ | | 1 |
| *Morganella morganii* | 1 (0.9) | _ | 1 | _ | | _ |
| *Achromobacter dentrificans* | 1 (0.9) | _ | _ | _ | 1 | _ |
| *Ewingella americana* | 1 (0.9) | _ | 1 | _ | _ | _ |
| *Sphingomonas sp.* | 1 (0.9) | 1 | _ | _ | _ | 1 |
| *Pantoea agglomerans* | 1 (0.9) | 1 | 1 | _ | _ | _ |
| **Total no. (%)** | 110 (55) | 34 (30.9) | 69 (62) | 18 (16) | 16 (14) | 7 (6) |

P value < 0.0086.

*Serratia marcescens*, *Enterobacter cloacae*, *Stenotrophomonas maltophilia*, *Morganella morganii*, *Achromobacter dentrificans*, *Sphingomonas sp*. and *Pantoea agglomerans*, and included one isolate each (n = 1, 1.5%). The predominant bacteria were *E. coli*, *K. pneumoniae* and *A. baumannii* (20, 30%, 15, 23%, 11, 16% strains, respectively).

It was found that females were more likely to acquire imipenem-resistant infections (60%) than males (40%). Infection rates were diverse among different age groups, with the maximum average being observed among patients of the 60–70 years age group. The isolates were predominantly isolated from urine (38/65), sputum (15/65), swabs (8/65) and blood (4/65) (Table 2).

## Antimicrobial susceptibility profile results

According to the Vitek-2 automated system test results, all 65 carbapenemase-producing isolates were multidrug-resistant. The susceptibility of the isolates to different antimicrobial agents revealed that almost all strains were highly resistant to beta-lactam antibiotics (piperacillin and cefazolin 96%, ceftriaxone 95%, cefepime 93% and ceftazidime 89%). Among the isolates, the percentages of resistance to imipenem, meropenem and ertapenem were 27 (41%), 26 (40%) and 24 (36%), respectively. Carbapenems resistance has been associated with resistance to other antibiotic classes. The rates of resistance to other antibiotics were: trimethoprim/sulphamethoxazole 75%; ciprofloxacin 67%; gentamicin 66%; nitrofurantoin 56%; piperacillin/tazobactam 55%; amikacin 49%; cefoxitin 36%; and tigecycline 32%. The isolated carbapenemase-producing Gram-negative bacilli exhibited a high significance difference in resistance to several tested antimicrobial agents (p < 0.05) (Table 2).

## Patterns of phenotypic resistance of carbapenemase-producing isolates

In general, almost all carbapenemase-producing strains were resistant to at least three classes of antimicrobials and were considered multidrug-resistant (MDR). In comparison, extensive drug-resistant (XDR) is classified as resistance to all but two or less classes of antimicrobials, and resistant to all classes of available antimicrobial agents is known as pan drug-resistant (PDR) [1, 26].

Our findings revealed that of 65 carbapenemase-producing Gram-negative bacilli isolates, 24 (36%) were defined as multidrug-resistant to 3 from 5 classes of antimicrobials. The most common phenotypic resistance patterns (β-lactams and β-lactamase inhibitor combinations, cephalosporins, quinolone and sulphonamides) were represented among the *E. coli* isolates, consisting of 4 classes of antimicrobials.

Furthermore, 17 (29%) of the recovered carbapenemase-producing isolates were extensively drug-resistant from 6 to 7 antimicrobial classes. The most common phenotypic resistance patterns were six (β-lactams and β-lactamase inhibitor combinations, cephalosporins, aminoglycosides, quinolone, sulphonamides and tetracyclines) among *Klebsiella sp*.

Unfortunately, 16 (24%) of isolates were PDR to all the tested antimicrobial classes (β-lactams and β-lactamase inhibitor combinations, cephalosporins, aminoglycosides, quinolone, sulphonamides, tetracyclines, nitrofurantoin and carbapenems) which were prevalent among *Acinetobacter baumannii*. Statistically, there was a significant difference in the distribution of MDR, XDR and PDR among carbapenemase-producing Gram-negative bacilli strains (*p* < 0.05) (Table 3).

## Phenotype detection of carbapenemase activity

The phenotype-based detection results of the Carba plus assay showed that 65 of 110 beta-lactams-resistant isolates of Gram-negative bacilli produced one or more carbapenemases,

**Table 2. Characteristic and antibiogram of 65 carbapenemase-producing Gram negative bacilli.**

| Bacteria | Total no. of isolates | Specimen no. and types | β-lactam and β-lactamase inhibitor combination | | Cephalosporins | | | | | Carbapenems | | | Aminoglycosides | | Quinolone | Nitrofurantoin | Sulphonamide | Tetracycline |
|---|---|---|---|---|---|---|---|---|---|---|---|---|---|---|---|---|---|---|
| | | | piperacillin | piperacillin-tazobactam | cefazoline | ceftriaxone | cefepime | ceftazidime | cefoxitin | imipenem | meropenem | ertapenem | amikacin | gentamicin | ciprofloxacin | nitrofurantoin | trimethoprim/Sulphamethoxazole | tigecycline |
| *E. coli* | 20 [30.7] | 17 urines, 2 bloods, 1 swab | 20 [100] | 10 [50] | 20 [100] | 20 [100] | 19 [95] | 20 [100] | 6 [30] | 1 [5] | 1 [5] | 1 [5] | 5 [25] | 8 [40] | 10 [50] | 3 [15] | 14 [70] | 3 [15] |
| *Klebsiella sp.* | 15 [23] | 12 urines, 2 sputum, 1 swab | 15 [100] | 6 [40] | 15 [100] | 15 [100] | 15 [100] | 15 [100] | 5 [33] | 4 [26] | 2 [13] | 2 [13] | 5 [33] | 10 [66] | 11 [73] | 8 [53] | 12 [80] | 3 [20] |
| *Acinetobacter baumanii* | 11 [16.9] | 1 urine, 1 swab, 9 sputum | 11 [100] | 11 [100] | 11 [100] | 11 [100] | 11 [100] | 11 [100] | 10 [90] | 11 [100] | 11 [100] | 11 [100] | 11 [100] | 11 [100] | 11 [100] | 11 [100] | 11 [100] | 9 [81] |
| *Pseudomonas aeruginosa* | 8 [12.3] | 5 urines, 2 swabs, 1 sputum | 8 [100] | 4 [50] | 8 [100] | 7 [87] | 8 [100] | 4 [50] | 0 | 4 [50] | 4 [50] | 4 [50] | 5 [62] | 6 [75] | 5 [62] | 7 [87] | 4 [50] | 2 [25] |
| *Proteus sp.* | 4 [6.1] | 1 urine, 1 swab, 2 sputum | 4 [100] | 2 [50] | 4 [100] | 4 [100] | 3 [75] | 3 [75] | 1 [25] | 3 [75] | 3 [75] | 2 [50] | 2 [50] | 1 [25] | 3 [75] | 3 [75] | 3 [75] | 1 [25] |
| *Serratia marcescens* | 1 [1.5] | sputum | 1 [100] | 0 | 1 [100] | 1 [100] | 1 [100] | 1 [100] | 0 | 1 [100] | 1 [100] | 1 [100] | 1 [100] | 1 [100] | 0 | 1 [100] | 1 [100] | 0 |
| *Enterobacter cloacae* | 1 [1.5] | swab | 1 [100] | 1 [100] | 1 [100] | 1 [100] | 1 [100] | 1 [100] | 1 [100] | 1 [100] | 1 [100] | 1 [100] | 1 [100] | 1 [100] | 1 [100] | 1 [100] | 1 [100] | 1 [100] |
| *Stenotrophomonas maltophilia* | 1 [1.5] | blood | 1 [100] | 1 [100] | 1 [100] | 1 [100] | 1 [100] | 1 [100] | 0 | 1 [100] | 1 [100] | 1 [100] | 1 [100] | 1 [100] | 1 [100] | 1 [100] | 1 [100] | 1 [100] |
| *Morganella morganii* | 1 [1.5] | urine | 0 | 0 | 1 [100] | 0 | 0 | 0 | 0 | 0 | 0 | 0 | 0 | 1 [100] | 1 [100] | 1 [100] | 1 [100] | 0 |
| *Achromobacter dentrificans* | 1 [1.5] | swab | 0 | 0 | 0 | 0 | 0 | 0 | 0 | 0 | 0 | 0 | 0 | 1 [100] | 0 | 1 [100] | 0 | 0 |
| *Sphingomonas sp.* | 1 [1.5] | blood | 1 [100] | 1 [100] | 1 [100] | 1 [100] | 1 [100] | 1 [100] | 0 | 1 [100] | 1 [100] | 1 [100] | 1 [100] | 1 [100] | 1 [100] | 0 | 1 [100] | 1 [100] |
| *Pantoea agglomerans* | 1 [1.5] | urine | 1 [100] | 0 | 0 | 1 [100] | - | - | 1 [100] | 0 | 1 [100] | 0 | 0 | 1 [100] | 0 | 0 | 0 | 0 |
| Total no. [%] | 65 | 38 urines, 15 sputum, 8 swabs, 4 bloods | 63 [96] | 36 [55] | 63 [96] | 62 [95] | 61 [93] | 58 [89] | 24 [36] | 27 [41] | 26 [40] | 24 [36] | 32 [49] | 43 [66] | 44 [67] | 37 [56] | 49 [75] | 20 [32] |

P value < 0.0001.

**Table 3. Distribution of MDR, XDR, and PDR patterns among 65 carbapenemase-producing Gram-negative bacilli strains.**

| Bacteria | Resistance profile | The most common phenotypic resistance patterns | Number of antimicrobial resistance classes |
|---|---|---|---|
| *E. coli* n = 20 | 13MDR | β-lactams and β-lactamase inhibitor combinations: Piperacillin and Piperacillin /Tazobactam | 3–4 |
| | | Cephalosporins: ceftazidime, ceftriaxone, cefazoline, cefepime and cefoxitin | |
| | | Sulphonamides: trimethoprim-sulphamethoxazole | |
| | | Quinolone: ciprofloxacin | |
| | 1XDR | β-lactams and β-lactamase inhibitor combinations: Piperacillin and Piperacillin/Tazobactam | 6 |
| | | Cephalosporins: ceftazidime, ceftriaxone, cefazoline, and cefepime | |
| | | Aminoglycosides: gentamicin and amikacin | |
| | | Quinolone: ciprofloxacin | |
| | | Sulphonamides: trimethoprim-sulphamethoxazole | |
| | | Tetracyclines: tigecycline | |
| | 1PDR | *β-lactams & β-lactamase inhibitor combinations: Piperacillin and Piperacillin/Tazobactam | 8 |
| | | Cephalosporins: ceftazidime, ceftriaxone, cefazoline, cefepime and cefoxitin | |
| | | Carbapenems: imipenem, ertapenem and meropenem | |
| | | Sulphonamides: trimethoprim-sulphamethoxazole | |
| | | Tetracyclines: tigecycline | |
| | | Aminoglycosides: gentamicin and amikacin | |
| | | Quinolone: ciprofloxacin | |
| | | Nitrofurantoin: (in urinary infection) | |
| *Klebsiella sp.* n = 15 | 5MDR | β-lactams: Piperacillin | 3–5 |
| | | Cephalosporins: ceftazidime, ceftriaxone, cefazoline, and cefepime | |
| | | Sulphonamides: trimethoprim-sulphamethoxazole | |
| | | Aminoglycosides: gentamicin ‘ | |
| | | Quinolone: ciprofloxacin | |
| | 6XDR | β-lactams and β-lactamase inhibitor combinations: Piperacillin and Piperacillin/Tazobactam | 6 |
| | | Cephalosporins: ceftazidime, ceftriaxone, cefazoline, cefepime and cefoxitin | |
| | | Sulphonamides: trimethoprim-sulphamethoxazole | |
| | | Aminoglycosides: gentamicin and amikacin | |
| | | Quinolone: ciprofloxacin | |
| | | Nitrofurantoin: (in urinary infection) | |
| | 2 PDR | * | 8 |
| *Acinetobacter baumanii* n = 11 | 2XDR | β-lactams and β-lactamase inhibitor combinations: Piperacillin and Piperacillin/Tazobactam | 7 |
| | | Cephalosporins: ceftazidime, ceftriaxone, cefazoline, cefepime and cefoxitin | |
| | | Carbapenems: imipenem, ertapenem and meropenem | |
| | | Sulphonamides: trimethoprim-sulphamethoxazole | |
| | | Aminoglycosides: gentamicin and amikacin | |
| | | Quinolone: ciprofloxacin | |
| | | Nitrofurantoin: (in urinary infection) | |
| | 9PDR | * | 8 |
| *Pseudomonas aeruginosa* n = 8 | 3MDR | β-lactams and β-lactamase inhibitor combinations: Piperacillin and Piperacillin/Tazobactam | 3–5 |
| | | Cephalosporins: ceftazidime, ceftriaxone, cefazoline, and cefepime | |
| | | Nitrofurantoin: (in urinary infection) | |
| | | Sulphonamides: trimethoprim-sulphamethoxazole | |

(*Continued*)

**Table 3.** (Continued)

| Bacteria | Resistance profile | The most common phenotypic resistance patterns | Number of antimicrobial resistance classes |
|---|---|---|---|
| | 4XDR | β-lactams and β-lactamase inhibitor combinations: Piperacillin and Piperacillin/Tazobactam | 6–7 |
| | | Cephalosporins: ceftazidime, ceftriaxone, cefazoline, cefepime and cefoxitin | |
| | | Carbapenems: imipenem, ertapenem and meropenem | |
| | | Nitrofurantoin: (in urinary infection) | |
| | | Aminoglycosides: gentamicin and amikacin | |
| | | Quinolone: ciprofloxacin | |
| | 1 PDR | * | 8 |
| *Proteus sp.* n = 4 | 1MDR | β-lactams and β-lactamase inhibitor combinations: Piperacillin and Piperacillin/Tazobactam | 3–4 |
| | | Cephalosporins: ceftriaxone, and cefazoline | |
| | | Carbapenems: imipenem, ertapenem and meropenem | |
| | | Quinolone: ciprofloxacin | |
| | 3XDR | β-lactams and β-lactamase inhibitor combinations: Piperacillin and Piperacillin/Tazobactam | 6–7 |
| | | Cephalosporins: ceftazidime, ceftriaxone, cefazoline, and cefepime | |
| | | Carbapenems: imipenem, ertapenem and meropenem | |
| | | Sulphonamides: trimethoprim-Sulphamethoxazole | |
| | | Aminoglycosides: gentamicin and amikacin | |
| | | Quinolone: ciprofloxacin | |
| | | Nitrofurantoin: (in urinary infection) | |
| *Serratia marcescens* n = 1 | 1XDR | β-lactams: Piperacillin | 6 |
| | | Cephalosporins: ceftazidime, ceftriaxone, cefazoline, and cefepime | |
| | | Carbapenems: imipenem, ertapenem and meropenem | |
| | | Sulphonamides: trimethoprim-Sulphamethoxazole | |
| | | Aminoglycosides: gentamicin and amikacin | |
| | | Nitrofurantoin: (in urinary infection) | |
| *Enterobacter* n = 1 | 1PDR | * | 8 |
| *Stenotrophomonas* n = 1 | 1PDR | * | 8 |
| *Morganella morganii* n = 1 | 1MDR | Cephalosporins: cefazoline | 5 |
| | | Sulphonamides: trimethoprim-sulphamethoxazole | |
| | | Tetracyclines: tigecycline | |
| | | Aminoglycosides: gentamicin and amikacin | |
| | | Quinolone: ciprofloxacin | |
| *Achromobacter* n = 1 | _ | _ | _ |
| *Sphingomonas sp.* n = 1 | 1PDR | * | 8 |
| *Pantoea agglomerans* n = 1 | 1MDR | β-lactams: Piperacillin | 4 |
| | | Carbapenems: imipenem, ertapenem and meropenem | |
| | | Cephalosporins: ceftazidime, ceftriaxone, cefazoline, and cefepime | |
| | | Aminoglycosides: gentamicin | |
| **Total no. (%)** | 24MDR | | |
| | 17XDR | | |
| | 16PDR | | |

* Most common phenotypic resistance patterns in relation to PDR are the same for all bacteria.

P value < 0.0182.

including 20 (30%) with Class A carbapenemase KPC (*E. coli*, n = 8; *Klebsiella sp.*, n = 5; *Pseudomonas aeruginosa*, n = 3; *Proteus sp.*, n = 2; *Acinetobacter baumanii.*, n = 1; and *Achromobacter dentrificans*, n = 1); 18 (27%) with Class B carbapenemase MBL (*Klebsiella sp.*, n = 5; *Acinetobacter baumanii.*, n = 9; *Pseudomonas aeruginosa* n = 1; and *Stenotrophomonas maltophilia; Serratia marcescens;* and *Proteus sp.*, n = 1, respectively); and 30 (46%) with Class D carbapenemase OXA-48 (*E. coli*, n = 8; *Klebsiella sp.*, n = 4; *Acinetobacter baumanii*, n = 9; *Pseudomonas aeruginosa*, n = 4; *Proteus sp.*, n = 2; and n = 1 for each of *Serratia marcescens; Sphingomonas sp.*; and *Pantoea agglomerans*). In addition, AmpC beta-lactamase production with porin loss was detected in 18 (27%) isolates (Table 4).

The highest rates of KPC and AmpC were found among *E. coli* isolates, while the maximum rates for MBL and OXA-48 were recorded among *A. baumanii* isolates. Detailed results of carbapenemase phenotype detection by Carba plus are shown in (Table 4). Statistically, there is a significant difference in the frequency of carbapenemase classes A, B and D with AmpC production among beta-lactam-resistant isolates of Gram-negative bacilli ($p < 0.0001$).

The results of the Carba plus test are revealed in (Fig 1). The positive result of KPC-producing isolates indicates that there is ≥5 mm difference only between disc C and disc A (<5 mm difference between disc B or disc D and disc A) (Fig 1A). The positive result of MBL-producing isolates shows that there is ≥5 mm difference in the disc zone diameters detected only between disc B and disc A (<5 mm difference between disc C or disc D and disc A) (Fig 1B). The positive result of the OXA-48-producing isolate displays that there is the absence of synergistic effects among discs A–D, inhibition zone diameter of ≤10 mm around disc E (Fig 1C). The positive result of AmpC with porin loss, shows that there is ≥5 mm difference in both discs D and C zone diameter compared with disc A (<5 mm difference between disc B and

**Table 4. Phenotype detection of carbapenemase-producing isolates of Gram-negative bacilli.**

| Bacteria | Tested isolates (%) | Carbapenemase Classes | | | AmpC + Porin loss (%) | Total no. of carbapenemase producing isolates (%) |
|---|---|---|---|---|---|---|
| | | Class A KPC (%) | Class B MBL (%) | Class D OXA-48 (%) | | |
| | | KPC | MBLs | OXA-48 | | |
| *E. coli* | 49(40) | 8(40) | _ | 8(26) | 8(44) | 20(30.7) |
| *Klebsiella sp.* | 23(20) | 5(25) | 5(27) | 4(13) | 4(22) | 15(23) |
| *Acinetobacter baumanii* | 13(11) | 1(5) | 9(50) | 9(30) | 1(5) | 11(16.9) |
| *Pseudomonas aeruginosa* | 10(9) | 3(15) | 1(5) | 4 (13) | 2(11) | 8(12.3) |
| *Proteus sp.* | 5(4) | 2(10) | 1(5) | 2 (6) | _ | 4(8.1) |
| *Serratia marcescens* | 3 (2) | _ | 1(5) | 1(3) | _ | 1(1.5) |
| *Pantoea agglomerans* | 1(0.9 | _ | _ | 1(3) | 1(5) | 1(1.5) |
| *Enterobacter cloacae* | 1(0.9) | _ | _ | _ | 1(5) | 1(1.5) |
| *Sphingomonas sp.* | 1(0.9) | _ | _ | 1(3) | _ | 1(1.5) |
| *Stenotrophomonas maltophilia* | 1(0.9) | _ | 1(5) | _ | _ | 1(1.5) |
| *Achromobacter dentrificans* | 1(0.9) | 1(5) | _ | _ | _ | 1(1.5) |
| *Ewingella americana* | 1(0.9) | _ | _ | _ | _ | _ |
| *Morganella morganii* | 1(0.9) | _ | _ | _ | 1 (5) | 1(1.5) |
| **Total no. (%)** | 110 | 20(30) | 18(27) | 30(46) | 18(27) | 65(59%) |

P value < 0.0182.

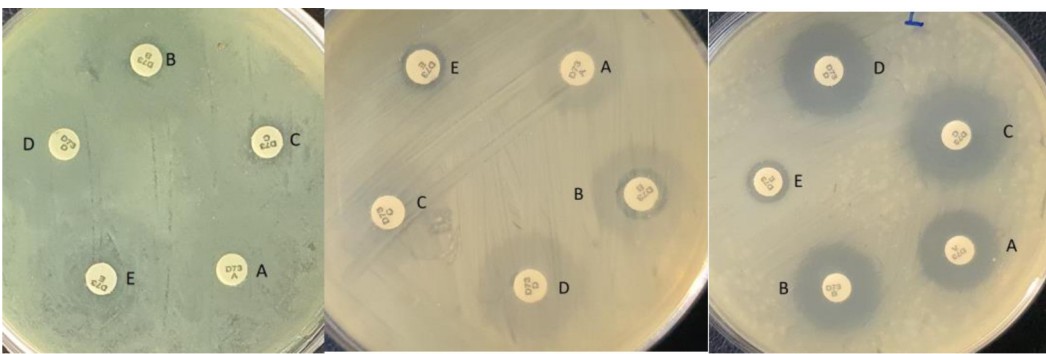

KPC positive (a)     MBL positive (b)     OXA-48 positive (c)

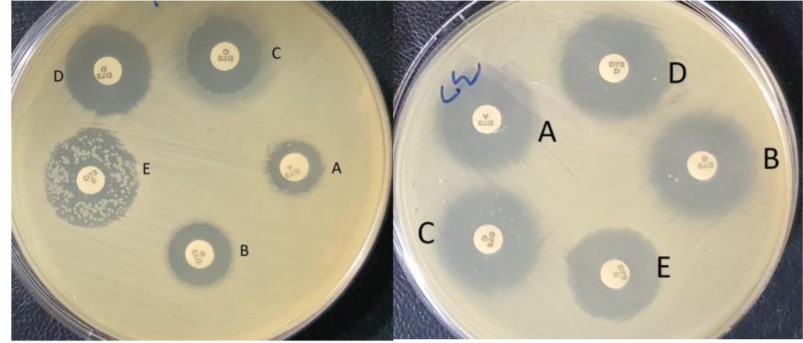

AmpC positive (d)     Negative result (e)

**Fig 1. Positive results for MBL, KPC, OXA-48, and AmpC + porin loss isolates for carbapenemase-producing, and a negative result for non-carbapenemase-producing Gram-negative bacilli isolates detected by Carba plus assay.** The difference in disc zone diameters was observed between disc A (faropenem), disc B (MBLs inhibitor), disc C (KPC inhibitor), disc D (AmpC inhibitor), and disc E (temocillin in combination with MBLs inhibitor). The positive result is shown for isolates that possess KPC, disc C- disc A ≥5 mm and the difference between each disc B- disc A and disc D- disc A <5 mm (a). The positive result of MBL having isolate shows that disc B- disc A ≥5 mm and difference between each disc C- disc A and disc D- disc A <5 mm (b). OXA-48 positive result showing that there is no increased zone on disc A- disc B, disc C, or disc D, respectively, and disc E ≤10 mm (c). The positive result of AmpC with porin loss shows that there is a ≥5 mm difference in both discs D and C zone diameter compared with disc A (<5 mm difference between disc B and disc A) (d). Negative results for non-carbapenemase-producing Gram-negative bacilli display that the diameter of the inhibition zone of all discs varies by ≤2 mm, and disc E > 10 mm (e).

disc A) (Fig 1D). In contrast, the negative results of isolates of non-carbapenemase-producing Gram-negative bacilli isolate display that disc A, B, C and D inhibition zone diameters vary by ≤2 mm, and the disc E >10 mm (Fig 1E). Thus, when looking at the PCR results, the MAST-Carba plus can correctly identify the 50/53 carbapenemase producers regardless of carbapenemase types.

## Prevalence of carbapenemase genes

After gene extraction, 53 phenotypically confirmed carbapenemase-producing isolates were examined genotypically based on the primers described previously for five carbapenemase genes KPC (bla KPC gene), MBLs (bla IMP, bla VIM and bla NDM genes) and OXA-48 (bla OXA-48 gene), by singleplex and multiplex PCR methods (Figs 2 and 3). Overall detection of genes, 51/53 (96%) of the isolates were positive for one or more carbapenemase genes. The result was including 50 (94%) isolates producing MBL (44 NDM producers, 26 VIM

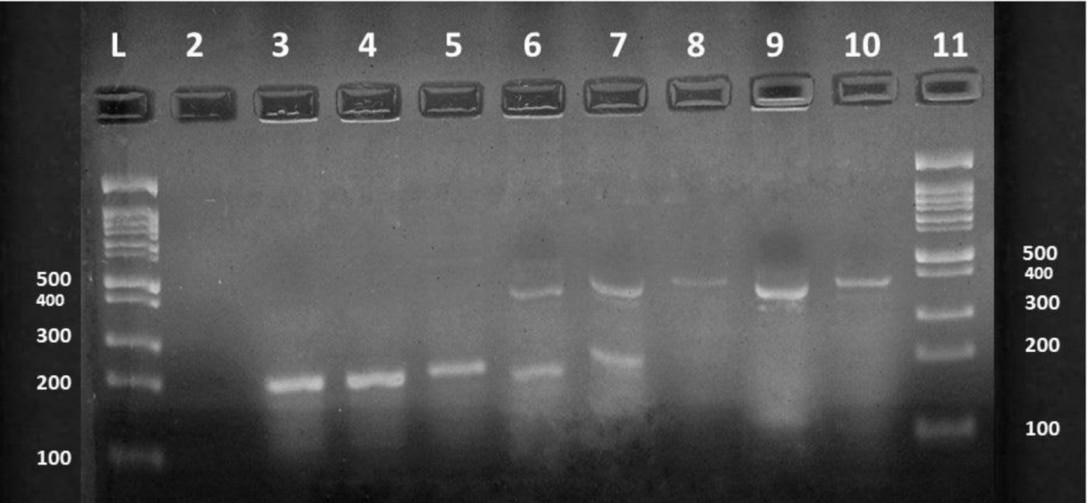

**Fig 2. Detection of carbapenemase genes IMP, and VIM in carbapenemase-producing Gram-negative bacilli isolates, using multiplex PCR.** Lanes 3–5 represent the positive IMP carbapenemase gene, Lanes 8–10 represent the positive VIM carbapenemase gene, Lanes 6 and 7 represent co-expression of IMP and VIM genes. The molecular size of the IMP and VIM genes are 232 and 390 bp respectively. Lane 1 and 11 are 1-kb DNA ladders. Lane 2 corresponds to the negative control.

producers, and 23 IMP producers), 40 (75%) isolates producing OXA-48 and 4 (7%) with KPC. MBL producers belong to twelve diverse Gram-negative bacilli isolates, the OXA-48 producers belong to ten different Gram-negative bacilli isolates, while the remaining four KPCs belong to three different Gram-negative bacilli isolates. In general, *Klebsiella sp*. 12 (22%) and *A. baumanii* 11 (20%), were the two species with the largest number of these genes, followed by *E. coli* 10 (18%), *Pseudomonas aeruginosa* 8 (15%), *Proteus sp*. 3 (5%) and one isolate (1%) for each of *Serratia marcescens*, *Stenotrophomonas malophilia*, *Sphingomonas sp*., *Achromobacter dentrificans*, *Enterobacter cloacae*, *Morganella morganii* and *Pantoea agglomerans*.

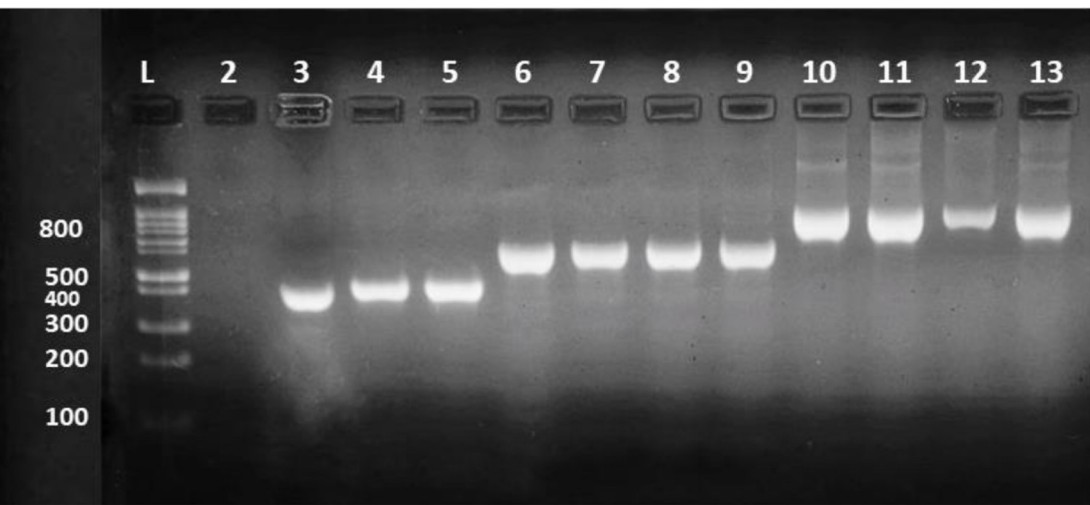

**Fig 3. Detection of the carbapenemase genes OXA, NDM, and KPC in carbapenemase-producing Gram-negative bacilli isolates, using singleplex PCR.** Lanes 3–5 represent the positive OXA-48 gene, Lanes 6–9 represent the positive NDM gene, and Lanes 10–13 represent the positive KPC gene. Lane 1 is a 1-kb DNA ladder. Lane 2 corresponds to the negative control. The molecular size of the tested *bla* OXA-48, *bla* NDM, and *bla* KPC carbapenemase genes are 438, 621 and 798 bp respectively.

Moreover, most of the carbapenemase-producing isolates produced more than one carbapenemase gene 46 (92%). General features of carbapenemase-producing isolates are shown in (Table 5). Statistical study verified that there is a highly significant difference in the prevalence of carbapenemase genes among the tested isolates ($p < 0.0001$).

## Discussion

Increased drug resistance is creating a real problem for treatment options, especially with high carbapenem resistance among Gram-negative bacilli isolates, which has been demonstrated and reported in many countries. As a result, the clinical effect of carbapenem resistance has become a serious public health crisis on a global scale [27]. Currently, the proliferation of carbapenemases is the most pressing resistance problem in Gram-negative bacteria.

Clinically important carbapenemase-producing pathogens are a triple risk as a result of their MDR profile, increased occurrence, and rapid distribution from species to species and even between different genera of Gram-negative bacteria through transmissible genetic elements (e.g., insertion sequences, transposons and plasmids). Furthermore, transmissible genetic elements harbour many additional antimicrobial resistance genes, subsequent in MDR or XDR traits, which greatly restricts treatment selection [18].

The microbiological examination in the present study revealed that 55% (110/200) broad-spectrum beta-lactam resistant isolates of Gram-negative bacilli were recovered from different clinical specimens. The most predominant bacterial pathogen retrieved was *E. coli* (44%), followed by *K. pneumoniae* (20%) and *A. baumannii* (11%). Statistical analysis showed that there was a significant difference in the prevalence of different bacterial pathogens and carbapenemase-resistant isolates recovered from different types of specimens ($p < 0.05$). The uppermost number of clinical isolates was collected from urine and sputum from the urinary tract infections and respiratory tract infections, respectively (Table 1). These results are in agreement

**Table 5. Genotype detection of carbapenemase-producing Gram-negative bacilli.**

| Bacteria | Tested isolates | KPC | MBL n = 50 | | | OXA-48 | Total genes no. (%) | Two or more genes |
|---|---|---|---|---|---|---|---|---|
| | | | NDM | IMP | VIM | | | |
| ***E. coli*** | 10(33) | 1 (10) | 8(80) | 5(50) | 4 (40) | 7 (70) | 10(18) | 8 |
| ***Klebsiella sp.*** | 14(26) | - | 10(71) | 1(7) | 11(78) | 8 (57) | 12(22) | 10 |
| ***Acinetobacter baumanii*** | 11(20) | 2 (18) | 9 (81) | 8(72) | 5 (45) | 10 (90) | 11(20) | 11 |
| *Pseudomonas aeruginosa* | 8(15) | - | 7 (87) | 3 (37) | 2 (25) | 7 (87) | 8(15) | 7 |
| *Proteus sp.* | 3(5) | - | 3(100) | 2 (66) | 2 (66) | 3 (100) | 3(5) | 3 |
| *Serratia marcescens* | 1(1.8) | - | 1(100) | 1 (100) | - | 1(100) | 1(1) | 1 |
| *Enterobacter cloacae* | 1(1.8) | - | 1(100) | - | - | 1 (100) | 1(1) | 1 |
| *Stenotrophomonas maltophilia* | 1(1.8) | - | 1(100) | 1 (100) | - | 1 (100) | 1(1) | 1 |
| *Morganella morganii* | 1(1.8) | - | 1(100) | - | 1 (100) | 1 (100) | 1(1) | 1 |
| *Achromobacter dentrificans* | 1(1.8) | 1 (100) | 1(100) | - | 1 (100) | 1 (100) | 1(1) | 1 |
| *Sphingomonas sp.* | 1(1.8) | - | 1(100) | 1 (100) | - | 1 (100) | 1(1) | 1 |
| *Pantoea agglomerans* | 1(1.8) | - | 1(100) | 1 (100) | - | 1 (100) | 1(1) | 1 |
| **Total no. (%)** | 53 | 4(7) | 44 (83) | 23(43) | 26 (49) | 40 (75) | 51(96) | 46(92) |

P value < 0.0001.

with those obtained by Jalalvand et al. and Rimrang et al. [7, 28] who demonstrated that the most predominant Gram-negative bacilli were *E. coli* and *K. pneumoniae*.

The detection rate of beta-lactam drug-resistant isolates of Gram-negative bacilli in clinical settings has gradually increased. This is possibly due to repeated exposure to drugs such as penicillin and 3rd generation cephalosporins. Similar resistance mechanisms have been detected in other studies where the loss of porin expression or ESBL enzymes in combination with AmpC beta-lactamases and expression of common efflux pumps, are a factor in the acquisition of carbapenemase resistance in Gram-negative pathogens [12].

Carbapenemases have been widely reported in *Enterobacteriaceae* worldwide [29]. However, limited studies of carbapenemase-producing Gram-negative bacilli have been conducted in the Kurdistan region of northern Iraq. Our study revolved around the dissemination of carbapenem-resistant Gram-negative bacilli bacteria, which provides information on the extent of the problem in the field from a microbiological and epidemiological point of view.

The frequency of carbapenem resistance in our community throughout the study period was 34/110 (30.9%) (Table 1). The isolation rates of carbapenem-resistant Gram-negative bacilli from various countries was found to be: 36% in Egypt [30], 13.6% and 37.9% in Iran [7, 26], respectively, 56% in Pakistan [31], 24.6% in China [27], 19% in Algeria [4], 2.82% in Turkey [19], 86.3% in Tunis [23], 5.99% in Morocco [29] and 2.9% in Ghana [12].

The most prevalent carbapenem-resistant isolated Gram-negative bacilli were *A. baumannii*, followed by *K. pneumoniae*, *P. aeruginosa* and *E. coli*. Conforming with other studies, the most common isolated species in this study was *A. baumanii* [4, 5].

The current emergence of carbapenemase-producing bacteria, especially *Enterobacteriaceae*, is of concern because it is often associated with the occurrence of multidrug-resistant isolates, where there are very few, if any, drug options available for them. Therefore, detection and initial identification of carbapenemase-producing bacteria are important [12].

Overall, 59% (65/110) of strains, including carbapenem-resistant strains of Gram-negative bacilli, were expressing phenotypic production of carbapenemases. The results showed that *E. coli* ranked first among Gram-negative bacteria, while *K. pneumoniae* ranked second in terms of the number of detected isolates (Table 2). Similar to data from Iran and Lebanon, the most common isolated species is *E. coli* and the second is *Klebsiella pneumoniae* [7, 24]. Urinary tract infections were found to be the dominant type of infection, 38 out of 65 isolates (58%) were from urine cultures, followed by respiratory tract infections (23%), wound infections (12%) and blood infections associated with systemic disease (6%). Urinary tract infections were particularly found to be associated with *E. coli* and *Klebsiella sp*. This finding is consistent with studies on invasive urinary tract infections documenting *E. coli* as a major pathogen among Gram-negative isolates [7, 14].

Concerning the antimicrobial susceptibility profiles (Table 2), the retrieved carbapenemase-producing isolates demonstrated the increased resistance level for penicillin as well as 1st, 3rd and 4th generation cephalosporins, sulphonamide, quinolones and aminoglycosides. The resistance rates for imipenem, meropenem and ertapenem were 41%, 40% and 36%, respectively. The progress of such resistant isolates is reflected in public health warnings.

Significant antimicrobial resistance was detected among carbapenem-resistant Gram-negative isolates compared to other MDR isolates. The resistance pattern of *A. baumanii* to imipenem in the current study (100%) is consistent with the reports of Boral et al. [32] but lower rates (85%, 77.5% and 61.89%) of imipenem resistance were reported by Shokri et al. [26], Shamim et al. [33] and Ain et al. [31], respectively. *Acinetobacter spp*. has been found to be a pathogen notorious for nosocomial infections. This is most often associated with wound infections and was the causative agent of 92% of infections associated with imipenem resistance [31]. Furthermore, the recovered carbapenemase-producing isolates were susceptible to tigecycline

which affected approximately 70% of MDR isolates. These results were in agreement with the observations of Shokri et al. [26].

Of interest, a high incidence of MDR, XDR and PDR profiles was observed among all carbapenemase-producing isolates. 65 isolates of the Gram-negative bacteria exhibited different patterns of resistance including 24 (36%) MDR, 17 (26%) XDR and 16 (24%) PDR patterns (Table 3).

The remarkable prevailing resistance regarding MDR to 4 classes, XDR to 6 classes and PDR to 8 classes of tested antimicrobials was observed among *E. coli*, *Klebsiella sp.* and *A. baumannii* isolates, respectively. This is supported by observations in the literature [26].

The higher rate of MDR pattern among the strains of Gram-negative bacteria carrying the carbapenemase gene in the current study, may be attributed to higher selection pressure due to self-medication, empirical use and overuse of carbapenems and 3rd generation cephalosporins, and a lack of careful monitoring of resident MDR isolates in hospital settings, particularly in the surgery ward, ICU and burns ward. For these reasons, more monitoring programs are needed in this area [34].

MDR and XDR isolates of these bacteria are increasingly being reported globally, but according to available results, higher rates than those previously reported were observed. A study conducted in Tehran Hospital in 2020 found that two types of bacteria, one *A. baumannii* and one of the *P. aeruginosa* strains were PDR [26].

The development of these PDR isolates is worrying as there is practically no antibiotic to treat them, and therefore morbidity and mortality rates due to infection with these strains are expected to be high. Some studies display high death rates (between 40 and 65%) among infected patients with *P. aeruginosa* and *Acinetobacter spp.* [35].

Among the 65 carbapenemase-producing Gram-negative isolates with a positive phenotype-based result, 30 (46%), 20 (30%), and 18 (27%) isolates were positive for OXA-48, KPC and MBL enzymes, respectively, or containing two of these enzymes, as well as the production of 27% of AmpC with porin loss (Table 4).

Despite the difference in detecting carbapenemase classes using the phenotypic method when compared with the genotyping method, in general, when looking at PCR results, Carba plus can correctly identify 51/53 (96%) carbapenemase-producers. It is interesting that through the application of the phenotyping method in the current study, it was demonstrated that almost all isolates of carbapenemase-producing Gram-negative bacilli selected for the genotyping method were able to produce different types of carbapenemases by the genotype-based method. It is important to note that this study points to Carba plus as potentially offering an effective in situ method for discriminating KPC, MBLs and OXA-48 carbapenemase genes in clinical *Enterobacteriaceae* isolates. This is ideal for application in the basic microbiology laboratory, providing real laboratory resolutions to help in the stewardship of antibiotics [10].

We have shown that Carba plus is appropriate for the detection of carbapenemase classes, which is comparable to previous results by Ohsaki et al. [21]. Carba plus displayed a brilliant discriminatory capability of carbapenemase types among multiple and single carbapenemase producers. Identification accuracy was found to be mostly high for IMP MBL producers, and for OXA-48 producers which may be difficult to detect due to reduced MICs to carbapenems [36].

To confirm the accuracy and dependability of identification results, we used molecular approaches. Reliable identification of carbapenemase genes is essential for disease outbreak detection. Out of 53 phenotypically confirmed carbapenemase-producing Gram-negative bacilli isolates, 51/53 (96%) isolates were harbouring one or more than one gene, while in two isolates none of the genes were detected. The most common carbapenemase gene was *bla*$_{NDM}$ 83% (44/53) followed by *bla*$_{OXA-48}$ 75% (40/53), *bla*$_{VIM}$ 49% (26/53) and *bla*$_{IMP}$ 43% (23/53), while the gene *bla*$_{KPC}$ was least frequent (4/53) (Table 5). Interestingly, *bla*$_{NDM}$ was the most

commonly detected carbapenemase and *bla* OXA-48 was second, which is consistent with other reports [12, 16, 19, 37]. In addition, similar to previous analyses, the two genes *bla* OXA-48 and *bla* NDM were commonly detected with a high predominance of *bla* OXA-48 [23, 26]. The New Delhi metallo-β-lactamase (*bla* NDM-1) gene was identified in New Delhi in an XDR *K. pneumoniae* in 2009 [5]. The gene is currently prevalent in other *Enterobacteriaceae*, *A. baumannii*, *K. oxytoca*, *P. aeruginosa*, *M. morganii*, *Citrobacter freundii* and *E. cloacae*, as well as in other countries including the United States and United Kingdom. This may be because NDM enzymes are encoded on highly mobile, conjugative plasmids which ease horizontal inter and intra-species transfer between bacteria rather than clonal spread [19].

We detected carbapenemases genes in different MDR Gram-negative bacilli bacteria. The most prevalent bacteria were *Klebsiella sp.*, *A. baumanii*, followed by *E. coli* and *P. aeruginosa*, which is similar to data from recent studies conducted in other countries [4, 23, 26, 29, 38]. Comparable to previous reports [14, 38], the KPC gene was the least detected among carbapenemase-producing bacteria, even in some studies none of *bla* KPC was detected [28, 37]. A probable explanation for the low detection rate of the KPC gene among carbapenemase-producing isolates might be that these isolates contain other class A carbapenemases (other than KPC), for example GES, Sme, NMC-A and IMI. This is backed up by reports elsewhere [39].

This result was in agreement with the observations of Mahrach et al. that the *bla* OXA-48 gene was detected in 75% of isolates [29]. According to the study by El-Badawi et al. the *bla* OXA-48 gene was detected in 73.68% of carbapenemase-producing isolates [39].

Finding OXA-48 producers is a major and worrisome issue. We observed a high frequency of infection with OXA-48 carbapenemase-producing isolates, the majority of OXA-48-producing isolates were *A. baumannii*, which is comparable to what was reported in a study by Bourafa et al. [4]. However, in contrast to our study, *K. pneumoniae* were the main OXA-48-producing isolates in other studies [12, 19].

Understanding the contribution of other factors, such as the production of ESBLs or defects in cell wall permeability, is necessary to increase the level of resistance to carbapenems and cephalosporins. In other studies, a close correlation was demonstrated between the expression of ESBLs (SHV and CTX-M-15) and OXA-48 enzyme production [31, 34]. Co-expression of AmpC enzymes and OXA-48 in carbapenem-resistant *Enterobacteriaceae* has previously been shown [19]. These results confirm the existence of other resistance mechanisms in our isolates.

OXA-48 is widely spread all over the world and has been a source of hospital infections globally. Of late, OXA-48 has been documented in a number of the Mediterranean nation-states as an epidemic, including in Turkey, Iran, Morocco, Lebanon and Tunisia [29].

This study revealed a high prevalence of MBL genes (*bla* VIM, *bla* IMP, and *bla* NDM), 50/53 (94%) of carbapenemase-producing isolates (Table 5) and all the three genes assayed were detected in the study samples. Based on the data from this study, there is a rapid emergence of MBL-producing isolates in some Erbil city hospitals. It appears that a similar pattern of distribution of the *bla* NDM gene may exist in other countries of the world [26, 31, 37, 38].

It is interesting that not all isolates producing carbapenemase are resistant to carbapenem. This is because carbapenemase production always raises the MICs of carbapenems, but may not be sufficiently high to be classified as resistant or intermediate resistant [20].

We observed a large number of isolates 92% (46/51) that were involved in the production of more than one carbapenemase gene (Table 5). The co-harbouring of carbapenemase isolates has been detected in several studies [19, 34]. It is worth noting that the coexistence of these carbapenemase genes is a therapeutic challenge to clinicians, due to restricted treatment options and the potential for global spread by horizontal transfer [40]. Interestingly, some isolates were found to contain two unrelated carbapenemase genes (*bla* OXA-48 and *bla* NDM) as previously observed in other countries [4, 5, 12, 38].

In the present work, isolates carrying 2 to 4 carbapenemase genes were commonly detected, especially among *Klebsiella sp*. and *A. baumannii*. This indicates that these microorganisms were able to accumulate numerous pan drug or multidrug-resistant determinants [34]. These isolates have already been described in other reports [5, 31, 34, 37].

Single isolates harbouring multiple genes have been described and identified as a key threat to antibacterial chemotherapy [34]. In accordance with the observations of Aruhomukama et al., Okoche et al., and Sadeghi et al. [8, 34, 38], the most important combinations were observed among *K. pneumoniae* isolates frequently carrying multiple and broad-spectrum beta-lactamases.

## Conclusions

The results of our study illustrate the emergence of carbapenemase producing Gram-negative bacilli isolates from patients at the referral hospitals in the Kurdistan Region of Iraq. The most common carbapenem-resistant bacteria that have been identified as responsible in healthcare-related infections are *A. baumannii*, *K. pneumoniae* and *E. coli*. The recovered clinical isolates demonstrated noteworthy multidrug resistance to penicillins, cephalosporins, sulphonamides, quinolones and aminoglycosides, which are considered a threat to public health. Retrieved isolates of carbapenemase-producing Gram-negative bacilli were markedly MDR, XDR or PDR to many classes of antimicrobials, a warning signal indicating complex therapy for diseases caused by these resistant microorganisms. Tigecycline showed promising antimicrobial activity against MDR, XDR and PDR Gram-negative bacilli. Carbapenem resistance by phenotypic detection is usually associated with production of classes D, A and B carbapenemases as well as AmpC production with porin loss.

Moreover, all carbapenem-resistant genes (*bla* $_{OXA}$, *bla* $_{IMP}$, *bla* $_{NDM}$, *bla* $_{VIM}$ and *bla* $_{KPC}$) were identified in the study specimen. The most prevalent gene was *bla* $_{NDM}$ and the lowest was *bla* $_{KPC}$.

The combination of phenotypic and genotypic detection of carbapenem-resistant isolates is an effective and reliable epidemiological process. Therefore, this study recommends continuous monitoring of antimicrobial susceptibility testing to detect multidrug-resistant isolates along with limited and appropriate use of antibiotics in hygienic practices. Accurate identification of carbapenem-resistant bacterial pathogens is essential in treating patients, as well as advancing appropriate contamination control measures to curb the rapid spread of these strains.

## Supporting information

**S1 Fig.**
(PDF)

**S1 Table. Characteristic and antibiogram profile of 65 carbapenemase-producing Gram-negative bacilli.**
(XLSX)

**S2 Table. The number of bacteria positive for carbapenemase genes and number of genes per organism n = 51.**
(XLSX)

## Acknowledgments

The authors would like to thank the management and staff at the Research Centre for their assistance.

## Author Contributions

**Data curation:** Sayran Hamad Haji.

**Formal analysis:** Sayran Hamad Haji, Fattma A. Ali.

**Investigation:** Sayran Hamad Haji.

**Methodology:** Sayran Hamad Haji, Safaa Toma Hanna Aka, Fattma A. Ali.

**Supervision:** Safaa Toma Hanna Aka, Fattma A. Ali.

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
