## [Decision Letter · Decision Letter 0]

23 Jul 2021

PONE-D-21-22590

Evaluation of carbapenemase-producing clinical isolates of Gram-negative bacilli using phenotypic and genotypic methods at hospitals in Erbil city, Kurdistan region of Iraq

PLOS ONE

Dear Dr. haji,

Thank you for submitting your manuscript to PLOS ONE. After careful consideration, we feel that it has merit but does not fully meet PLOS ONE’s publication criteria as it currently stands. Therefore, we invite you to submit a revised version of the manuscript that addresses the points raised during the review process.

ACADEMIC EDITOR: A major revision is needed.The manuscript should be revised for English editing and grammar mistakes.

We look forward to receiving your revised manuscript.

Kind regards,

Abdelazeem Mohamed Algammal, Prof, Ph.D

Academic Editor

PLOS ONE

Journal Requirements:

2. In your Methods section, please provide additional information on the verbal consent provided by the patients, including as to how this verbal consent was documented or witnessed.

5. Please ensure that you refer to Figure 1 in your text as, if accepted, production will need this reference to link the reader to the figure.

6. We note you have included a table to which you do not refer in the text of your manuscript. Please ensure that you refer to Table 1 in your text; if accepted, production will need this reference to link the reader to the Table.

Reviewers' comments:

Reviewer's Responses to Questions

**Comments to the Author**

1. Is the manuscript technically sound, and do the data support the conclusions?

Reviewer #1: Partly

Reviewer #2: Yes

2. Has the statistical analysis been performed appropriately and rigorously? 

Reviewer #1: No

Reviewer #2: No

3. Have the authors made all data underlying the findings in their manuscript fully available?

Reviewer #1: Yes

Reviewer #2: Yes

4. Is the manuscript presented in an intelligible fashion and written in standard English?

Reviewer #1: Yes

Reviewer #2: Yes

5. Review Comments to the Author

Reviewer #1: Comments to authors:

- The current study is very interesting; however, the authors should address the following comments to improve the quality of the manuscript:

- The manuscript should be revised for language editing and grammar mistakes by a native English speaker.

- Please write the genes name in correct form all over the manuscript (the 1st 3 letters must be small and italic and the other 3 letter are capital and subscripted)

Title:

I think the work would benefit from the title that contains main conclusion of the study (should be derived from the conclusion), please modify the title.

Abstract:

- The abstract must illustrates the used methods and the most prevalent results (give more hints about methods and results). Besides, rephrase the main conclusion of your findings.

Introduction:

-Give a hint about different infections caused by common pathogenic members of Enterobacteriaceae (such as E. coli and K. pneumonia) and Pseudomonas aeruginosa, their virulence factors, and the mechanism of disease occurrence.

- The introduction needs to be more informative:

-The authors should illustrate the public heath importance concerning the emergence of multidrug-resistant (MDR) bacterial pathogens that reflecting the necessary of new potent and safe antimicrobial agents. Several studies proved the widespread MDR- bacterial pathogens;

Authors could add the following paragraph:

Multidrug resistance has been increased globally that is considered public health threat. Several recent studies reported the existence of multidrug-resistant bacterial pathogens from different origins including humans, poultry, cattle, and fish that increase the need for routine application of the antimicrobial susceptibility testing to detect the antibiotic of choice as well as screening of the emerging MDR strains. You should cite the following valuable studies:

1-PMID: 33177849

2-PMID: 32497922

3-PMID:33061472

4-PMID: 33947875

5-PMID: 32472209

6-PMID: 32994450

-Rephrase the aim of work to be clear and better sound.

Material and methods

-Table 1 must be placed in the results section.

-It is not acceptable to illustrate the prevalence of the bacterial pathogens in the methods; it should be placed to the results section.

-Where are the methods of the isolation and identification of 200 Gram-negative bacteria?? You must illustrate in details and support your methods with specific references.

- Phenotype-based method for carbapenemases: should be modified to be: Phenotypedetection carbapenemases using Disc diffusion method. Rephrase this section; add more details, discuss in details and specific references.

- Confirmation of carbapenemase production using commercial

combination disk assay; add specific references to this section.

-Molecular detection of genes should be modified:

PCR-based detection of carbapenemase genes. Besides, add specific references.

-Where are the statistical analyses?

-Results:

- Where are the results of phenotypic characters and the prevalence of the isolated bacterial pathogens from the examined samples? You must illustrate their prevalence in a new table.

-Table 1 must be placed in the results section.

- Illustrate in a new table the test results of the phenotypic MDR (illustrate the antimicrobial classes of the used antimicrobial agents).

-Illustrate the distribution of the carbapenemase genes among the recovered isolates in a new table.

- Please increase the resolution of all figures (1200 dpi).

-Where are the statistical analyses?

-Discussion:

-The discussion is too long and poor; the authors are advised to illustrate the real impact of their findings without repetition of results.

-Conclusion

- Should be rephrased to be sounded. A real conclusion should focus on the question or claim you articulated in your study, whose resolution has been the main objective of your paper?

Reviewer #2: - The current study has a significant impact, but it needs a major revision:

- The manuscript should be revised for grammar mistakes.

-The title is broad, please modify the title.

-In the introduction: discuss the public health importance of the members of Enterobacteriaceae and other Gram negative bacterial pathogens.

-Improve the aim of work.

Methods:

-Transfer Table 1 to the results.

-Transfer the prevalence of bacterial species to the results section.

-Discuss the isolation and identification of the bacterial pathogens.

-Specific references should be added to all the used methods and techniques.

-Where is the statistical analysis?

-Results:

- Discuss the prevalence of the retrieved isolates in a new table.

- Transfer Table 1 to the results

- Please increase the resolution of all figures (1200 dpi).

-Where is the statistical analysis?

-Discussion:

-Very long; please improve

6. PLOS authors have the option to publish the peer review history of their article (what does this mean?). If published, this will include your full peer review and any attached files.

Reviewer #1: No

Reviewer #2: No

---

## [Author Response · Author response to Decision Letter 0]

4 Sep 2021

Dear Dr. Abdelazeem 

Thank you for reviewing my manuscript. I appreciate your efforts and the efforts of the reviewers.

I have worked hard to edit the revised manuscript and I hope I have done well

I look forward to having my research accepted in PLOS ONE journal.

Sayran H. Haji

Author 

Journal Requirements:

I tried to be in accordance with the requirements of PLOS ONE style

2. In your Methods section, please provide additional information on the verbal consent provided by the patients, including as to how this verbal consent was documented or witnessed. 

I added regarding oral consent in the ethical issues section

My cover letter contains the DOI of the original, uncropped, unedited images of the blot /gel data published in a public data repository.

https://data.mendeley.com/datasets/fhnkrf3jdr/1

Upon re-submitting your revised manuscript, please upload your study’s minimal underlying data set as either Supporting Information files or to a stable, public repository and include, or accession numbers within your revised cover letter. For a list of acceptable repositories, please see http://journals.plos.org/plosone/s/data-availability#loc-recommended-repositories. Any potentially identifying patient information must be fully anonymized.

The minimal data set is available in public repository and include the relevant DOIs, within my revised cover letter. 

Dataset: https://data.mendeley.com/datasets/dp7g8rntyb/1

I don’t have any ethical or legal restrictions

5. Please ensure that you refer to Figure 1 in your text as, if accepted, production will need this reference to link the reader to the figure.

I referenced all the Figures in my text to link the reader to the figure.

6. We note you have included a table to which you do not refer in the text of your manuscript. Please ensure that you refer to Table 1 in your text; if accepted, production will need this reference to link the reader to the Table.

I referenced all the tables in my script to connect the reader to the figure.

Reviewer #1: Comments to authors:

- The current study is very interesting; however, the authors should address the following comments to improve the quality of the manuscript:

- The manuscript should be revised for language editing and grammar mistakes by a native English speaker.

I have checked for language editing and grammatical errors by native British speakers using online proofreading;

Professional editing and proofreading

services at your fingertips

https://app.proofreadmyessay.co.uk/my-documents/

- Please write the genes name in correct form all over the manuscript (the 1st 3 letters must be small and italic and the other 3 letter are capital and subscripted)

I corrected the name of the genes throughout the manuscript

Title:

I think the work would benefit from the title that contains main conclusion of the study (should be derived from the conclusion), please modify the title.

I modified the title that contains main conclusion

Abstract:

- The abstract must illustrates the used methods and the most prevalent results (give more hints about methods and results). Besides, rephrase the main conclusion of your findings.

More hints were given about the methods and results. Besides, I reformulated the main conclusion of the results.

Introduction:

-Give a hint about different infections caused by common pathogenic members of Enterobacteriaceae (such as E. coli and K. pneumonia) and Pseudomonas aeruginosa, their virulence factors, and the mechanism of disease occurrence.

Hints were given about the common pathogenic members of Enterobacteriaceae in the introduction part

- The introduction needs to be more informative:

-The authors should illustrate the public heath importance concerning the emergence of multidrug-resistant (MDR) bacterial pathogens that reflecting the necessary of new potent and safe antimicrobial agents. Several studies proved the widespread MDR- bacterial pathogens;

Authors could add the following paragraph:

Multidrug resistance has been increased globally that is considered public health threat. Several recent studies reported the existence of multidrug-resistant bacterial pathogens from different origins including humans, poultry, cattle, and fish that increase the need for routine application of the antimicrobial susceptibility testing to detect the antibiotic of choice as well as screening of the emerging MDR strains. You should cite the following valuable studies:

1-PMID: 33177849

2-PMID: 32497922

3-PMID:33061472

4-PMID: 33947875

5-PMID: 32472209

6-PMID: 32994450

I have illustrated the public health importance of the emergence of multidrug-resistant (MDR) bacterial pathogens by adding your paragraph, and I really appreciate your help in providing these valuable studies, which I used as references.

-Rephrase the aim of work to be clear and better sound. 

I paraphrased the aim of the work

Material and methods

-Table 1 must be placed in the results section. 

I have moved the table to the resulting part

-It is not acceptable to illustrate the prevalence of the bacterial pathogens in the methods; it should be placed to the results section. I have moved the prevalence of bacterial pathogens to the results section.

-Where are the methods of the isolation and identification of 200 Gram-negative bacteria?? You must illustrate in details and support your methods with specific references.

The methods for isolating and identifying 200 Gram-negative bacteria have been explained with adding specific references. Furthermore, I added a new table to identify 200 Gram-negative bacteria.

- Phenotype-based method for carbapenemases: should be modified to be: Phenotypedetection carbapenemases using Disc diffusion method. Rephrase this section; add more details, discuss in details and specific references.

I have modified the phenotypic-based method for carbapenemes to the phenotype detection of carbapenemes using the disc diffusion method, and more details have been added to this section with references.

- Confirmation of carbapenemase production using commercial

combination disk assay; add specific references to this section. 

The phenotypic-based method has been combined with confirmation of carbapenemase production using commercial combination disk checked, because I think it was duplicated, and specific references were added

-Molecular detection of genes should be modified:

PCR-based detection of carbapenemase genes. Besides, add specific references.

Molecular gene detection was changed to PCR-based detection of carbapenemase genes. Besides, specific references have been added.

-Where are the statistical analyses? 

Statistical analyzes for this study were designed using GraphPad Prism (version 5; GraphPad Software, San Diego, CA). The Chi-square test was applied to analyze the obtained data.

-Results:

- Where are the results of phenotypic characters and the prevalence of the isolated bacterial pathogens from the examined samples? You must illustrate their prevalence in a new table.

The results of the phenotypic characteristics and prevalence of bacterial pathogens isolated from the examined samples were added to a new table with a clear explanation of the correlation in the text. ------------------------------------------------------------------------------------

-Table 1 must be placed in the results section. 

Table 1 has been converted to Table 2 and moved to the result section.

- Illustrate in a new table the test results of the phenotypic MDR (illustrate the antimicrobial classes of the used antimicrobial agents). 

I have developed a new table of MDR phenotypic test results identifying the antimicrobial classes of the antimicrobial agents used.

-Illustrate the distribution of the carbapenemase genes among the recovered isolates in a new table. 

The distribution of carbapenemase genes among the recovered isolates is illustrated in a new table

- Please increase the resolution of all figures (1200 dpi). 

I have increased the resolution. Hope it is clear now

-Where are the statistical analyses?

Statistical analyzes for this study were designed using GraphPad Prism (version 5; GraphPad Software, San Diego, CA). The Chi-square test was applied to analyze the obtained data.

-Discussion:

-The discussion is too long and poor; the authors are advised to illustrate the real impact of their findings without repetition of results. 

I've shortened the discussion and deleted all the duplicates as much as I can

-Conclusion

- Should be rephrased to be sounded. A real conclusion should focus on the question or claim you articulated in your study, whose resolution has been the main objective of your paper? 

The conclusion has been reformulated to be presented and includes the main resolution of my study.

Reviewer #2: - The current study has a significant impact, but it needs a major revision:

- The manuscript should be revised for grammar mistakes. 

I have checked for language editing and grammatical errors by native British speakers using online proofreading;

Professional editing and proofreading

services at your fingertips

https://app.proofreadmyessay.co.uk/my-documents/

-The title is broad, please modify the title. 

I have modified and shortened the title 

-In the introduction: discuss the public health importance of the members of Enterobacteriaceae and other Gram-negative bacterial pathogens. 

The public health importance of the members of Enterobacteriaceae and other Gram-negative bacterial pathogens was discussed in the introduction part

-Improve the aim of work. 

The purpose of the work has improved

Methods:

-Transfer Table 1 to the results.

 I have moved the table to the resulting part

-Transfer the prevalence of bacterial species to the results section. 

I have moved the prevalence of bacterial pathogens to the results section.

-Discuss the isolation and identification of the bacterial pathogens.

 The methods for isolating and identifying Gram-negative bacteria have been explained with adding specific references. Furthermore, I added a new table to identify 200 Gram-negative bacteria.

-Specific references should be added to all the used methods and techniques. 

I added specific references to all the methods and techniques used

-Where is the statistical analysis?

Statistical analyzes for this study were designed using GraphPad Prism (version 5; GraphPad Software, San Diego, CA). The Chi-square test was applied to analyze the obtained data.

-Results:

- Discuss the prevalence of the retrieved isolates in a new table.

The results of the prevalence of bacterial pathogens isolated from the examined samples were added to a new table with a clear explanation in the text.

- Transfer Table 1 to the results 

Table 1 has been converted to Table 2 and moved to the result section.

- Please increase the resolution of all figures (1200 dpi). 

I have increased the resolution. Hope it is clear now

-Discussion:

-Very long; please improve

I've shortened and improved the discussion as much as I can________________________________________

---

## [Decision Letter · Decision Letter 1]

7 Sep 2021

PONE-D-21-22590R1Prevalence and characterisation of carbapenemase encoding genes in multidrug-resistant Gram-negative bacilliPLOS ONE

Dear Dr. haji,

Thank you for submitting your manuscript to PLOS ONE. After careful consideration, we feel that it has merit but does not fully meet PLOS ONE’s publication criteria as it currently stands. Therefore, we invite you to submit a revised version of the manuscript that addresses the points raised during the review process.

We look forward to receiving your revised manuscript.

Kind regards,

Abdelazeem Mohamed Algammal, Prof, Ph.D

Academic Editor

PLOS ONE

Reviewers' comments:

Reviewer's Responses to Questions

**Comments to the Author**

1. If the authors have adequately addressed your comments raised in a previous round of review and you feel that this manuscript is now acceptable for publication, you may indicate that here to bypass the “Comments to the Author” section, enter your conflict of interest statement in the “Confidential to Editor” section, and submit your "Accept" recommendation.

Reviewer #1: (No Response)

2. Is the manuscript technically sound, and do the data support the conclusions?

Reviewer #1: Yes

3. Has the statistical analysis been performed appropriately and rigorously? 

Reviewer #1: Yes

4. Have the authors made all data underlying the findings in their manuscript fully available?

Reviewer #1: Yes

5. Is the manuscript presented in an intelligible fashion and written in standard English?

Reviewer #1: Yes

6. Review Comments to the Author

Reviewer #1: The authors have addressed some of my comments; however the have

ignored a major comment concerning the introduction.

Please address the following comment:

The introduction needs to be more informative:

-The authors should illustrate the public heath importance concerning the emergence

of multidrug-resistant (MDR) bacterial pathogens that reflecting the necessary of new

potent and safe antimicrobial agents. Several studies proved the widespread MDRbacterial pathogens;

Authors could add the following paragraph:

Multidrug resistance has been increased globally that is considered public health

threat. Several recent studies reported the existence of multidrug-resistant bacterial

pathogens from different origins including humans, poultry, cattle, and fish that

increase the need for routine application of the antimicrobial susceptibility testing to

detect the antibiotic of choice as well as screening of the emerging MDR strains. You

should cite the following valuable studies:

1-PMID: 33177849

2-PMID: 32497922

3-PMID:33061472

4-PMID: 33947875

5-PMID: 32472209

6-PMID: 32994450

7. PLOS authors have the option to publish the peer review history of their article (what does this mean?). If published, this will include your full peer review and any attached files.

Reviewer #1: No

---

## [Author Response · Author response to Decision Letter 1]

11 Sep 2021

Journal Requirements:

I tried to be in accordance with the requirements of PLOS ONE style

2. In your Methods section, please provide additional information on the verbal consent provided by the patients, including as to how this verbal consent was documented or witnessed. 

I added regarding oral consent in the ethical issues section

My cover letter contains the DOI of the original, uncropped, unedited images of the blot /gel data published in a public data repository.

https://data.mendeley.com/datasets/fhnkrf3jdr/1

Upon re-submitting your revised manuscript, please upload your study’s minimal underlying data set as either Supporting Information files or to a stable, public repository and include, or accession numbers within your revised cover letter. For a list of acceptable repositories, please see http://journals.plos.org/plosone/s/data-availability#loc-recommended-repositories. Any potentially identifying patient information must be fully anonymized.

The minimal data set is available in public repository and include the relevant DOIs, within my revised cover letter. 

Dataset: https://data.mendeley.com/datasets/dp7g8rntyb/1

I don’t have any ethical or legal restrictions

5. Please ensure that you refer to Figure 1 in your text as, if accepted, production will need this reference to link the reader to the figure.

I referenced all the Figures in my text to link the reader to the figure.

6. We note you have included a table to which you do not refer in the text of your manuscript. Please ensure that you refer to Table 1 in your text; if accepted, production will need this reference to link the reader to the Table.

I referenced all the tables in my script to connect the reader to the figure.

Reviewer #1: Comments to authors:

- The current study is very interesting; however, the authors should address the following comments to improve the quality of the manuscript:

- The manuscript should be revised for language editing and grammar mistakes by a native English speaker.

I have checked for language editing and grammatical errors by native British speakers using online proofreading;

Professional editing and proofreading

services at your fingertips

https://app.proofreadmyessay.co.uk/my-documents/

- Please write the genes name in correct form all over the manuscript (the 1st 3 letters must be small and italic and the other 3 letter are capital and subscripted)

I corrected the name of the genes throughout the manuscript

Title:

I think the work would benefit from the title that contains main conclusion of the study (should be derived from the conclusion), please modify the title.

I modified the title that contains main conclusion

Abstract:

- The abstract must illustrates the used methods and the most prevalent results (give more hints about methods and results). Besides, rephrase the main conclusion of your findings.

More hints were given about the methods and results. Besides, I reformulated the main conclusion of the results.

Introduction:

-Give a hint about different infections caused by common pathogenic members of Enterobacteriaceae (such as E. coli and K. pneumonia) and Pseudomonas aeruginosa, their virulence factors, and the mechanism of disease occurrence.

Hints were given about the common pathogenic members of Enterobacteriaceae in the introduction part

- The introduction needs to be more informative:

-The authors should illustrate the public heath importance concerning the emergence of multidrug-resistant (MDR) bacterial pathogens that reflecting the necessary of new potent and safe antimicrobial agents. Several studies proved the widespread MDR- bacterial pathogens;

Authors could add the following paragraph:

Multidrug resistance has been increased globally that is considered public health threat. Several recent studies reported the existence of multidrug-resistant bacterial pathogens from different origins including humans, poultry, cattle, and fish that increase the need for routine application of the antimicrobial susceptibility testing to detect the antibiotic of choice as well as screening of the emerging MDR strains. You should cite the following valuable studies:

1-PMID: 33177849

2-PMID: 32497922

3-PMID:33061472

4-PMID: 33947875

5-PMID: 32472209

6-PMID: 32994450

I have illustrated the public health importance of the emergence of multidrug-resistant (MDR) bacterial pathogens by adding your paragraph, and I really appreciate your help in providing these valuable studies, which I used as references.

-Rephrase the aim of work to be clear and better sound. 

I paraphrased the aim of the work

Material and methods

-Table 1 must be placed in the results section. 

I have moved the table to the resulting part

-It is not acceptable to illustrate the prevalence of the bacterial pathogens in the methods; it should be placed to the results section. I have moved the prevalence of bacterial pathogens to the results section.

-Where are the methods of the isolation and identification of 200 Gram-negative bacteria?? You must illustrate in details and support your methods with specific references.

The methods for isolating and identifying 200 Gram-negative bacteria have been explained with adding specific references. Furthermore, I added a new table to identify 200 Gram-negative bacteria.

- Phenotype-based method for carbapenemases: should be modified to be: Phenotypedetection carbapenemases using Disc diffusion method. Rephrase this section; add more details, discuss in details and specific references.

I have modified the phenotypic-based method for carbapenemes to the phenotype detection of carbapenemes using the disc diffusion method, and more details have been added to this section with references.

- Confirmation of carbapenemase production using commercial

combination disk assay; add specific references to this section. 

The phenotypic-based method has been combined with confirmation of carbapenemase production using commercial combination disk checked, because I think it was duplicated, and specific references were added

-Molecular detection of genes should be modified:

PCR-based detection of carbapenemase genes. Besides, add specific references.

Molecular gene detection was changed to PCR-based detection of carbapenemase genes. Besides, specific references have been added.

-Where are the statistical analyses? 

Statistical analyzes for this study were designed using GraphPad Prism (version 5; GraphPad Software, San Diego, CA). The Chi-square test was applied to analyze the obtained data.

-Results:

- Where are the results of phenotypic characters and the prevalence of the isolated bacterial pathogens from the examined samples? You must illustrate their prevalence in a new table.

The results of the phenotypic characteristics and prevalence of bacterial pathogens isolated from the examined samples were added to a new table with a clear explanation of the correlation in the text. ------------------------------------------------------------------------------------

-Table 1 must be placed in the results section. 

Table 1 has been converted to Table 2 and moved to the result section.

- Illustrate in a new table the test results of the phenotypic MDR (illustrate the antimicrobial classes of the used antimicrobial agents). 

I have developed a new table of MDR phenotypic test results identifying the antimicrobial classes of the antimicrobial agents used.

-Illustrate the distribution of the carbapenemase genes among the recovered isolates in a new table. 

The distribution of carbapenemase genes among the recovered isolates is illustrated in a new table

- Please increase the resolution of all figures (1200 dpi). 

I have increased the resolution. Hope it is clear now

-Where are the statistical analyses?

Statistical analyzes for this study were designed using GraphPad Prism (version 5; GraphPad Software, San Diego, CA). The Chi-square test was applied to analyze the obtained data.

-Discussion:

-The discussion is too long and poor; the authors are advised to illustrate the real impact of their findings without repetition of results. 

I've shortened the discussion and deleted all the duplicates as much as I can

-Conclusion

- Should be rephrased to be sounded. A real conclusion should focus on the question or claim you articulated in your study, whose resolution has been the main objective of your paper? 

The conclusion has been reformulated to be presented and includes the main resolution of my study.

Reviewer #2: - The current study has a significant impact, but it needs a major revision:

- The manuscript should be revised for grammar mistakes. 

I have checked for language editing and grammatical errors by native British speakers using online proofreading;

Professional editing and proofreading

services at your fingertips

https://app.proofreadmyessay.co.uk/my-documents/

-The title is broad, please modify the title. 

I have modified and shortened the title 

-In the introduction: discuss the public health importance of the members of Enterobacteriaceae and other Gram-negative bacterial pathogens. 

The public health importance of the members of Enterobacteriaceae and other Gram-negative bacterial pathogens was discussed in the introduction part

-Improve the aim of work. 

The purpose of the work has improved

Methods:

-Transfer Table 1 to the results.

 I have moved the table to the resulting part

-Transfer the prevalence of bacterial species to the results section. 

I have moved the prevalence of bacterial pathogens to the results section.

-Discuss the isolation and identification of the bacterial pathogens.

 The methods for isolating and identifying Gram-negative bacteria have been explained with adding specific references. Furthermore, I added a new table to identify 200 Gram-negative bacteria.

-Specific references should be added to all the used methods and techniques. 

I added specific references to all the methods and techniques used

-Where is the statistical analysis?

Statistical analyzes for this study were designed using GraphPad Prism (version 5; GraphPad Software, San Diego, CA). The Chi-square test was applied to analyze the obtained data.

-Results:

- Discuss the prevalence of the retrieved isolates in a new table.

The results of the prevalence of bacterial pathogens isolated from the examined samples were added to a new table with a clear explanation in the text.

- Transfer Table 1 to the results 

Table 1 has been converted to Table 2 and moved to the result section.

- Please increase the resolution of all figures (1200 dpi). 

I have increased the resolution. Hope it is clear now

-Discussion:

-Very long; please improve

I've shortened and improved the discussion as much as I can

---

## [Decision Letter · Decision Letter 2]

29 Sep 2021

PONE-D-21-22590R2Prevalence and characterisation of carbapenemase encoding genes in multidrug-resistant Gram-negative bacilliPLOS ONE

Dear Dr. haji,

Thank you for submitting your manuscript to PLOS ONE. After careful consideration, we feel that it has merit but does not fully meet PLOS ONE’s publication criteria as it currently stands. Therefore, we invite you to submit a revised version of the manuscript that addresses the points raised during the review process.

ACADEMIC EDITOR: Please address all the reviewer comments, it is your last chance./>==============================

We look forward to receiving your revised manuscript.

Kind regards,

Abdelazeem Mohamed Algammal, Prof, Ph.D

Academic Editor

PLOS ONE

Reviewers' comments:

Reviewer's Responses to Questions

**Comments to the Author**

1. If the authors have adequately addressed your comments raised in a previous round of review and you feel that this manuscript is now acceptable for publication, you may indicate that here to bypass the “Comments to the Author” section, enter your conflict of interest statement in the “Confidential to Editor” section, and submit your "Accept" recommendation.

Reviewer #1: (No Response)

2. Is the manuscript technically sound, and do the data support the conclusions?

Reviewer #1: Yes

3. Has the statistical analysis been performed appropriately and rigorously? 

Reviewer #1: Yes

4. Have the authors made all data underlying the findings in their manuscript fully available?

Reviewer #1: Yes

5. Is the manuscript presented in an intelligible fashion and written in standard English?

Reviewer #1: Yes

6. Review Comments to the Author

Reviewer #1: In the introduction section the authors reported several recent studies in the following paragraph,

however they have cited only one reference.. How??

Please cite all references:

Multidrug resistance has been increased globally that is considered public health

threat. Several recent studies reported the existence of multidrug-resistant bacterial

pathogens from different origins including humans, poultry, cattle, and fish that

increase the need for routine application of the antimicrobial susceptibility testing to

detect the antibiotic of choice as well as screening of the emerging MDR strains. You

should cite the following valuable studies:

1-PMID: 33177849

2-PMID: 32497922

3-PMID:33061472

4-PMID: 33947875

5-PMID: 32472209

6-PMID: 32994450

Lines 438- 505; please add specific references. The following sentence is found instead of refrences; please revise and correct.

Error! Reference source not found..

7. PLOS authors have the option to publish the peer review history of their article (what does this mean?). If published, this will include your full peer review and any attached files.

Reviewer #1: No

---

## [Author Response · Author response to Decision Letter 2]

7 Oct 2021

Reviewer #1: In the introduction section the authors reported several recent studies in the following paragraph,

however they have cited only one reference.. How??

Please cite all references:

With my appreciation, I have referred to all of the following valuable studies as references in the introduction, material methods, and results in the manuscript.

1-PMID: 33177849 reference number 30

2-PMID:33061472 reference number 3

3-PMID: 33947875 reference number 1 

4-PMID: 32472209 reference number 22

5-PMID: 32994450 reference number 2

Lines 438- 505; please add specific references. The following sentence is found instead of refrences; please revise and correct. Error! Reference source not found. 

I have revised and corrected the error related to the reference source on lines 438-505.

---

## [Decision Letter · Decision Letter 3]

11 Oct 2021

Prevalence and characterisation of carbapenemase encoding genes in multidrug-resistant Gram-negative bacilli

PONE-D-21-22590R3

Dear Dr. haji,

We’re pleased to inform you that your manuscript has been judged scientifically suitable for publication and will be formally accepted for publication once it meets all outstanding technical requirements.

Kind regards,

Abdelazeem Mohamed Algammal, Prof, Ph.D

Academic Editor

PLOS ONE

Additional Editor Comments (optional):

Reviewers' comments:

Reviewer's Responses to Questions

**Comments to the Author**

1. If the authors have adequately addressed your comments raised in a previous round of review and you feel that this manuscript is now acceptable for publication, you may indicate that here to bypass the “Comments to the Author” section, enter your conflict of interest statement in the “Confidential to Editor” section, and submit your "Accept" recommendation.

Reviewer #1: All comments have been addressed

2. Is the manuscript technically sound, and do the data support the conclusions?

Reviewer #1: Yes

3. Has the statistical analysis been performed appropriately and rigorously? 

Reviewer #1: Yes

4. Have the authors made all data underlying the findings in their manuscript fully available?

Reviewer #1: Yes

5. Is the manuscript presented in an intelligible fashion and written in standard English?

Reviewer #1: Yes

6. Review Comments to the Author

Reviewer #1: The authors have carried out a significant changes to the manuscript. They have addressed all the suggested corrections and comments. Really, it's an interesting study that has a significant impact. Now, the manuscript could be accepted.

Congratulations.

7. PLOS authors have the option to publish the peer review history of their article (what does this mean?). If published, this will include your full peer review and any attached files.

Reviewer #1: No

---

## [Editor Report · Acceptance letter]

21 Oct 2021

PONE-D-21-22590R3 

Prevalence and characterisation of carbapenemase encoding genes in multidrug-resistant Gram-negative bacilli 

Dear Dr. Haji:

I'm pleased to inform you that your manuscript has been deemed suitable for publication in PLOS ONE. Congratulations! Your manuscript is now with our production department. 

Kind regards, 

on behalf of

Professor Abdelazeem Mohamed Algammal 

Academic Editor

PLOS ONE